# Polymeric Nanomicelles Loaded with Anandamide and Their Renal Effects as a Therapeutic Alternative for Hypertension Treatment by Passive Targeting

**DOI:** 10.3390/pharmaceutics15010176

**Published:** 2023-01-03

**Authors:** Virna Margarita Martín Giménez, Marcela Analía Moretton, Diego Andrés Chiappetta, María Jimena Salgueiro, Miguel Walter Fornés, Walter Manucha

**Affiliations:** 1Instituto de Investigaciones en Ciencias Químicas, Facultad de Ciencias Químicas y Tecnológicas, Universidad Católica de Cuyo, San Juan J5400, Argentina; 2Cátedra de Tecnología Farmacéutica I, Facultad de Farmacia y Bioquímica, Universidad de Buenos Aires, Buenos Aires C1053, Argentina; 3Consejo Nacional de Investigaciones Científicas y Técnicas (CONICET), Godoy Cruz M2290, Argentina; 4Departamento de Física, Facultad de Farmacia y Bioquímica, Universidad de Buenos Aires, Buenos Aires C1053, Argentina; 5Instituto de Histología y Embriología “Dr. Mario H. Burgos” (IHEM), Consejo Nacional de Investigaciones Científicas y Técnicas (CONICET), Mendoza M5500, Argentina; 6Laboratorio de Farmacología Experimental Básica y Traslacional, Área de Farmacología, Departamento de Patología, Facultad de Ciencias Médicas, Universidad Nacional de Cuyo, Mendoza M5500, Argentina; 7Instituto de Medicina y Biología Experimental de Cuyo, Consejo Nacional de Investigación Científica y Tecnológica (IMBECU-CONICET), Mendoza M5500, Argentina

**Keywords:** anandamide, nanomicelles, hypertension, diuretic, natriuretic, kidneys

## Abstract

We have previously demonstrated significant in vitro natriuretic effects of anandamide (AEA) nanoformulation in polymeric nanoparticles, whose size prevents their accumulation in organs, such as the kidneys. Therefore, it is of particular interest to design and test nanostructures that can pharmacologically accumulate in these organs. In this regard, we prepared and characterized polymeric nanomicelles (~14 and 40 nm). Likewise, their biodistribution was determined. Spontaneously hypertensive rats (SHR) and normotensive rats (WKY), n = 3 per group, were divided into five treatment conditions: control, sham, free AEA freshly dispersed in aqueous solution or 24 h after its dispersion, and AEA encapsulated in nanomicelles. The kidneys were the main site of accumulation of the nanoformulation after 24 h. Freshly dispersed free AEA showed its classical triphasic response in SHR, which was absent from all other treatments. Nanoformulated AEA produced a sustained antihypertensive effect over 2 h, accompanied by a significant increase in fractional sodium excretion (FSE %). These effects were not observed in WKY, sham, or free AEA-treated rats after 24 h of its aqueous dispersion. Without precedent, we demonstrate in vivo natriuretic, diuretic, and hypotensive effects of AEA nanoformulation in polymeric nanomicelles, suggesting its possible use as a new antihypertensive agent with intravenous administration and passive renal accumulation.

## 1. Introduction

Hypertension is one of the leading causes of morbidity and mortality worldwide [1], where patients with renal diseases have a substantial risk of suffering hypertension, and vice versa [2]. Moreover, it is known that extracellular fluid volume regulation through renal sodium excretion plays a key role in maintaining blood pressure homeostasis. In fact, natriuresis is an exceptionally powerful mechanism of long-term blood pressure stabilization around a reference value in patients with hypertension [3]. Many therapeutic options have been developed for managing hypertension; however, they are insufficient to stop or significantly reduce the progression of this and other associated cardiovascular diseases. Additionally, they may produce undesirable side effects due to their unspecific biodistribution rather than specific drug localization in therapeutic targets of interest, such as the kidneys. Therefore, there is a great need to explore new technologies to overcome traditional therapeutic options’ limitations. Of particular interest, nanotechnology has become an attractive and novel solution to overcome many of the disadvantages of current cardiovascular therapies—mainly those related to drug targeting [4]. Drug delivery to the kidneys through renal targeting is an attractive but little-explored field. Just a few nanocarriers have been assessed for their localization at the renal level (including some polymeric and metal nanoparticles, dendrimers, liposomes, and solid lipid nanoparticles) using, in most cases, strategies of active targeting, such as surface functionalization, which is usually quite complex compared to other methods [5].

In this regard, the contribution of nanotechnology would be beneficial in the case of non-therapeutically exploited molecules, such as Anandamide (AEA)—which is an endocannabinoid that was discovered in 1992 and studied for the first time in 1996 as an active principle with antihypertensive effects [6]. Particularly, AEA exhibits many limitations when used in its free form, such as very poor water solubility, low bioavailability, physicochemical and enzymatic instability, adsorption onto plastic and glass surfaces, and side effects at the central nervous system (CNS) level [7]. Nevertheless, AEA nanoencapsulation with therapeutic aims has been scarcely assessed so far [7,8,9], and some of our previous studies suggested antihypertensive responses of this active principle through their actions at the renal level [10,11]. Although these trials have not been performed in vivo, our in vitro findings indicated possible diuretic and natriuretic effects (by inducing the expression of iNOS/NO levels and decreasing the Na^+^/K^+^ ATPase activity in HK2 cells) promoted by nanoformulated AEA in polymeric nanoparticles obtained through an electrospraying technique. This new pharmaceutical formulation has overcome most of the physicochemical limitations and disadvantages inherent to this active substance (mainly related to the maintenance of this endocannabinoid in its active form at different temperatures and for a long time), which, so far, has dismissed its therapeutic use [11]. Although polymeric nanoparticles have multiple benefits as nanocarriers for controlled drug delivery, their sizes often complicate their penetration and accumulation in different tissues as well as limit their route of administration [12]. The particle size is critical when the nanoformulation must arrive at organs with several filtration barriers, such as the kidneys [13].

Thus, it is a priority to search for another kind of nanostructure that is capable of being located at the renal level. An attractive option would be using polymeric nanomicelles to achieve this purpose. These carriers are nanosized vehicles, with diameters ranging from about 5 and 100 nm, which are composed of amphiphilic polymers and may be self-assembled in an aqueous medium above their critical micelle concentration. Polymeric micelles have a hydrophobic core and a hydrophilic corona. This spatial distribution confers the ability to encapsulate multiple lipophilic drugs to polymeric micelles, which bestows colloidal stability to this nanostructure [14,15]. This encapsulation allows for considerable improvement in the stability and solubility of the encapsulated drugs, which directly influences the enhancement of their cellular uptake and biological effects [16,17,18].

It is essential to choose an adequate polymer when fabricating nanomicelles (especially for renal uptake) since they should be biocompatible, biodegradable, and non-toxic [19]. In this regard, some of the amphiphilic polymers that are most often used to prepare nanomicelles with these features are the derivatives of poly(ethylene oxide)–poly(propylene oxide) block copolymers. They are usually classified into two commercialized groups: (i) the linear and bifunctional triblocks of PEO-PPO-PEO or poloxamers (Pluronic^®^) and (ii) their four-armed, branched counterparts, poloxamines (Tetronic^®^) [20]. Moreover, another well-researched amphiphilic biopolymer is D-α-Tocopheryl polyethylene glycol 1000 succinate (TPGS). It is a vitamin E hydrosoluble derivative associated with polyethylene glycol (PEG) 1000 that is authorized by the Food and Drug Administration (FDA) to be marketed as a drug solubilizer for topical, oral, and parenteral formulations because of its micellization properties [21].

Many times, changes in temperature and other factors may affect the stability of polymeric micelles. For this reason, it is necessary to evaluate possible alterations in the nanostructures under different conditions [22]. In addition, another relevant topic in nanotechnological pharmaceutical formulations is that minor differences—for example, in their surface charge or morphology—could significantly impact their activity, cellular accumulation, and biodistribution in the human body, which leads to diverse biological responses [23].

In this context, the aim of this study was, firstly, to develop and characterize both TPGS and PF127 polymeric micelles loaded with AEA, as well as to assess the biological distribution and renal accumulation through the passive targeting of each type of micelle. Secondly, we evaluated (especially at the renal level) the in vivo effect of these nanoformulations in an animal model of hypertension—comparing, at the same time, the pharmacokinetic and pharmacodynamic behavior of both kinds of micelles in order to know which would be the better option for the development of a new antihypertensive therapy.

## 2. Materials and Methods

### 2.1. Micellar Preparation and Characterization

For the preparation and characterization of D-α-tocopheryl polyethylene glycol succinate (TPGS) and Pluronic^®^ F127 (PF127) micelles, 30 mg of commercial TPGS ( Eastman Chemical Company, Kingsport, TN, USA) and PF127 (BASF, CABA, Buenos Aires, Argentina) were weighed and dissolved in 1 mL of miliQ water (Sigma-Aldrich, St. Louis, MO, USA), with continuous agitation at room temperature (25 °C) until their homogeneous dispersion. Then, 750 µg of AEA, dissolved in absolute ethanol (15 µL of commercial AEA ethanolic solution (Cayman Chemical, Ann Arbor, MI, USA), was added to the polymeric dispersion, maintaining agitation until the complete evaporation of ethanol was achieved. AEA-free polymeric micelles were used as controls.

The average hydrodynamic diameter (Dh), zeta potential (ZP), and polydispersity index (PDI) of the TPGS and PF127 micelles (3% *w*/*v*) (in the absence and presence of AEA) were determined through dynamic light scattering (DLS) (scattering angle of θ = 173 ° to the incident beam, Zetasizer Nano-ZSP, ZEN 5600, Malvern Instruments, Malvern, UK) at 25 and 37 °C. Samples were equilibrated for 5 min at 25 °C before measurements were taken (n = 3).

The size and morphology of the TPGS and PF127 empty micelles (3% *w*/*v*) were characterized through transmission electron microscopy (Zeiss 902, Zeiss, Oberkochen, Germany). Briefly, a droplet of each micellar dispersion (1–2 mg/mL) was placed onto a grid and covered with Fomvar (Sigma-Aldrich, St. Louis, MO, USA) film, stained with 5 μL of phosphotungstic acid solution (1% *w*/*v*), washed with distilled water (5 μL), and air-dried. Micrographs were obtained using a Gatan camera (CCD camera Orius SC 1000-11 Mpx, (Gatan, Inc., Pleasanton, CA, USA). The DigitalMicrograph software (version 3.4, Gatan, Inc., Pleasanton, CA, USA) defined the particle diameters.

### 2.2. Radiolabeling of TPGS and PF127 Micellar Systems

Labeling was performed by means of the direct method using ^99m^TcO_4_^−^, obtained from a radionuclide generator (Radiofarma^®^, BACON S.A.I.C., Villa Martelli, Buenos Aires, Argentina), and ClSn2, as a reducing agent (1 mg/mL), prepared in an acidic medium. For the procedure, 1 mL of the empty micellar system and 150 µL of Sn2+ were added, and the pH was corrected to neutral using NaOH to, finally, incorporate ^99m^TcO_4_^−^ from the generator. The preparation was incubated at room temperature for 20 min.

### 2.3. Radiochemical Purity Control

Radiochemical purity control was performed through thin-plate ascending chromatography using ITLC as the stationary phase and acetone as the mobile phase to show the ^99m^TcO_4_^−^ that might have remained free. Moreover, the ITLC stationary phase and the pyridine: acetic acid: water (5:3:1:5) mobile phase were used to highlight the hydrolyzed ^99m^Tc that might have been produced during labeling.

### 2.4. Stability of Labeling

This was performed by incubating the labeled preparation (~100 µL) in rat plasma (~1.2 mL) at 37 °C, from which serial samples were taken at 1, 3, and 24 h to verify that the efficiency of dialing and ^99m^TcO_4_^−^ was not detected. Verification was performed using the same methodology described in the previous point.

### 2.5. Biodistribution of Micellar Systems

#### 2.5.1. Administration

Approximately 0.2 mL of the labeled preparation was administered intravenously (lateral tail vein), taking into account the relationship with the body weight of female Sprague-Dawley rats (150–170 g). Each dose was measured before and after administration to record the net administered activity for subsequent calculations to determine the percentage of uptake for each organ.

#### 2.5.2. Gamma Camera Scintigraphic Images

Images were acquired in a small field planar gamma camera equipped with a high-resolution collimator and software dedicated to laboratory animals. During the acquisitions, the animals were anesthetized with 2% isofluorane in an O_2_ vehicle. The images obtained are presented in two color scales for the qualitative analysis of the distribution profile and capture of the systems.

#### 2.5.3. Ex Vivo Procedure

The euthanasia of the rats was completed, and subsequently, the organs of interest were removed. Then, they were washed with saline solution and weighed using an analytical balance. Each organ was measured in a calibrated solid scintillation counter. The results obtained were corrected by physical decay and are expressed as a percentage of the administered activity and as a percentage of the administered activity per gram of tissue.

### 2.6. In Vivo Biological Activity Studies

Adult male rats, WKY/Crl and SHR/Crl of 250–300 g, were purchased from the Faculty of Veterinary Sciences of the National University of La Plata (Charles Rivers Laboratories, Wilmington, MA, USA). Animals were housed in metal cages under controlled temperature and humidity conditions, with food and water ad libitum, and exposed to a cycle of 12 h of light and 12 h of darkness. The animals from both strains (n = 21 per group) were divided into 7 groups: control (basal), non-encapsulated AEA freshly dispersed in aqueous solution, non-encapsulated AEA 24 h after its preparation in aqueous solution, AEA/TPGS micelles, AEA/PF127 micelles, TPGS empty micelles, and PF127 empty micelles.

The rats were anesthetized through the intraperitoneal injection of ketamine-xylazine (80 mg/10 mg/Kg). To allow for spontaneous ventilation, a tracheotomy was performed. The left femoral vein was catheterized (PP25, Portex, Portland, UK) to infuse both saline solution at 10 mL/h (for the hydration of the animals) and the different treatments (bolus, at a dose of 100 µg of AEA/100 g of animal weight, or their equivalents in empty micelles). We determined the application dose by considering an encapsulation efficiency level close to 100%, since AEA (due to its highly lipophilic nature) is practically insoluble in aqueous media, as we mentioned before. Then, the amount of non-encapsulated AEA would be negligible, considering its intrinsic water solubility. The left femoral artery was cannulated for mean arterial pressure (MAP) monitoring. A pulled catheter was inserted into the urinary bladder to collect urine from both kidneys in an Eppendorf and measure urinary volume, electrolyte levels, creatinine concentrations, and osmolarity both in basal conditions and after the injection of the different treatments. Both the MAP measurement and urine collection were performed at different time points (basal, 30 min, 60 min, 90 min, and 120 min) after the administration of each treatment. Likewise, a blood sample was taken from the left femoral artery at the same time points to assess plasmatic osmolarity, creatinine concentrations, and electrolyte levels. Finally, with these values, we calculated the fractional excretion of sodium percentage (FES%) to know, more accurately, the amount of salt excreted by the kidneys of these animals over time in response to the different treatments. At the end of the experiments, all animals were slaughtered through exsanguination.

### 2.7. Statistical Analysis

Results are provided as the mean ± the standard error of the mean. Comparisons between groups were assessed using two-way analysis of variance (ANOVA II) analyses followed by Duncan’s test. A *p* < 0.05 was considered to be significant.

## 3. Results

### 3.1. Characterization of the Micellar Systems

The morphology of both micellar systems was spherical, as Figure 1 shows. DLS graphs (Figure 2A,B) and TEM images (Figure 1) allowed for the observation of the differences between the sizes of TPGS and PF127 micelles, which are around 14 and 40 nm, respectively. In addition, the DLS analysis of free AEA freshly dispersed in an aqueous solution (Figure 2C) showed the presence of aggregates up to 400 nm in diameter, which were not observed in the DLS graphs of the AEA/TPGS and AEA/PF127 micelles, thus confirming the complete encapsulation of AEA within these polymeric nanocarriers.

For both types of micelles (with and without AEA), Table 1 shows the values of ZP (at 25 °C) and Dh and PDI (at 25 °C and 37 °C), which demonstrates their thermal stability both at room and body temperature. In this regard, a DLS analysis showed that, at room temperature (25 °C), TPGS empty micelles showed a narrow and unimodal size distribution with, a Dh of 10.9 ± 0.1 and a PDI value of 0.032 ± 0.013. After the encapsulation of AEA within the micelles, the TPGS micelles population remained unimodal, with a Dh of 13.0 ± 0.4 and a PDI value of 0.234 ± 0.038 and observing a slight increase in micellar size. On the other hand, PF127 empty micelles showed a bimodal size distribution, with a smaller peak corresponding to 8.7% of the population with a Dh of 5.1 ± 0.5 and a greater peak corresponding to 91.3% of the population with a Dh of 35.4 ± 3.1 and a PDI value of 0.391 ± 0.041. After the encapsulation of AEA within the micelles, the PF127 micelles population remained bimodal—with a smaller peak corresponding to 7.2% of the population with a Dh of 5.8 ± 0.7 and a greater peak corresponding to 92.8% of the population with a Dh of 40.6 ± 1.5—with a PDI value of 0.299 ± 0.039, and while observing a little increase in micellar size. Likewise, at body temperature (37 °C), TPGS micelles without AEA showed a unimodal size distribution, with a Dh of 11.2 ± 0.1 and a PDI value of 0.020 ± 0.008. After the AEA encapsulation, the TPGS micelles population stayed unimodal, with a Dh of 13.4 ± 0.1 and a PDI value of 0.157 ± 0.008, and again, observing a small increase in micellar size. Otherwise, for PF127 empty micelles, the bimodal population disappeared and was replaced by only a unimodal population, with a Dh of 22.0 ± 1.2 and a PDI value of 0.154 ± 0.039. After the encapsulation of AEA within the micelles, the PF127 micelles population remained unimodal, with a Dh of 31.2± 0.3, a PDI value of 0.287 ± 0.007, and observing a 1.4-fold inccrease in micellar size.

### 3.2. Micellar Biodistribution

The resulting labels for TPGS 3% and PF127 3% yielded radiochemical purities higher than 90%, which were stable in plasma at 37 °C until 24 h of incubation.

Figure 3 shows the distribution of the injection of TPGS 3% micelles in the first minute after their administration (grouped 10 s per frame). The micelles could be detected from the tail vein (the bolus of the preparation is shown in green); then, the preparation ascended through the vena cava towards the heart (displayed in green) and was subsequently distributed to the rest of the organs (the body of the animal in blue). There was predominance in high-perfusion organs, such as the liver (the abdominal area begins to be visualized with a light green color, taking the shape of the liver superimposed on the kidneys, because it is a planar image with a superposition of structures in signal registration). Towards the end of the minute, the injection site was almost no longer observed, there was still a system circulating in the body, and abdominal uptake was maintained.

Figure 3 also shows the biological distribution of TPGS 3% at 30 min, 3 h, and 24 h after administration. At 30 min (Figure 3A), there was a high abdominal signal, mainly due to hepatic uptake. The kidneys captured behind the liver without clear delimitation, but bladder uptake confirmed renal elimination. There was marked system circulation (the whole silhouette of the animal appears in blue, showing the circulation). At 3 h (Figure 3B), there was a predominant abdominal signal due to liver uptake. The kidneys appeared in the lower area of the abdominal signal, as both renal silhouettes delimited below the hepatic uptake could be seen. Bladder accumulation was less than earlier. Intestinal uptake appeared (with an appreciable signal, since the intestine was closer to the detector, resulting in less attenuation). The circulation of the system was smaller, but the visible signal continued. At 24 h (Figure 3C), the circulating agent was no longer visible in the image, so the different organs had already taken up most of the system. Hepatic uptake was dominant, and there was also renal uptake (one of the renal silhouettes on the left is visible in the image). Finally, the intestinal uptake was no longer distinguished.

On the other hand, Figure 4 shows the first minute of the biological distribution of PF127 3% micelles after their administration. Abdominal (hepatic) uptake, from the beginning of the distribution, was observed. It is essential to remember that the liver is also a highly perfused organ, whereby part of the signal comes from the “overlap” of liver tissue uptake (phagocytosis of the system) and perfusion of the organ. On the other hand, this organ is very close to the detector, so the signal is not attenuated by other structures interposed between the liver and the detector.

Likewise, at 30 min (Figure 4A), the biodistribution pattern of PF127 3% was quite similar to that of the TPGS 3% micelles. There was a circulation of the marked system (the whole silhouette of the animal appears in blue, showing the circulation and also the highly perfused blood-filled organs, such as the heart and joints). At 3 h (Figure 4B), the circulation of the system persisted (with the figure showing the silhouette still marked and thoracic uptake in the heart and joints). There was high hepatic uptake, with visualization of the lower part of the renal silhouettes, and a slight intestinal uptake. At 24 h (Figure 4C), the circulating system persisted, liver uptake was predominant, and intestinal uptake was mild and seemed to have traveled.

Figure 5 shows the percentage of activity injected per gram of tissue, in each organ, and at each time analyzed.

### 3.3. Mean Arterial Pressure Measurement, Plasmatic and Urinary Parameters Assessment, and Fractional Excretion of Sodium Determination

The treatment with AEA freshly dispersed in an aqueous solution showed, in SHR, the classic triphasic response of free AEA (an initial phase characterized by a rapid and short drop in blood pressure, a subsequent phase that consists of brief pressor response, and a more prolonged final phase characterized by a considerable decrease in blood pressure), as Figure 6 shows. Notably, a triphasic response was not observed in SHR animals treated with AEA/TPGS, AEA/PF127, or free AEA after 24 h of its dispersion in an aqueous solution.

For its part, Figure 7 shows the MAP, and Figure 8 shows the FES%, of WKY and SHR rats in response to the different treatments at the time points studied. In the case of TPGS and PF127 micelles loaded with AEA, a slow but blunt reduction in the MAP was observed over 2 h (37 ± 2 mmHg with TPGS and 20 ± 1 mmHg with PF127), accompanied by a progressive and significant increase in FES% (0.45 ± 0.08% with TPGS and 0.43 ± 0.08% with PF127) and urinary volume (40 ± 10 μL/min with TPGS and 45 ± 8 μL/min with PF127). It is worth noting that free AEA freshly dispersed in an aqueous solution showed an increase in FES% of 0.41± 0.12% (slightly lower than that of AEA/TPGS and AEA/PF127 micelles) that occurred within the first 30 min but remained constant until the end of the assay 2 h later. However, unlike what happened with free AEA freshly dispersed in an aqueous solution, the increase in FES%, by both AEA/TPGS and AEA/PF127 micelles, was much more gradual during the 2 h span (Figure 8).

On the other hand, and as we expected, no significant decreases in MAP or increases in diuresis/natriuresis were observed in animals treated with empty micelles of TPGS and PF127 or free AEA after 24 h of its dispersion in an aqueous solution, confirming that the observed hypotensive effects were attributed exclusively to the nanoencapsulated AEA. Finally, none of the effects of nanoformulation or free AEA in SHR rats was observed in WKY rats.

## 4. Discussion

The present study demonstrates that AEA loaded into polymeric micelles may cause in vivo natriuretic and diuretic effects through passive renal targeting, thus favoring reductions in the blood pressure values of hypertensive animals. This finding constitutes a significant contribution to the field of precision medicine for developing more efficient and safer antihypertensive therapies, especially considering the lack of nanoformulations targeted to the kidneys for the treatment of hypertension.

Micellar size represents a key parameter for drug carrier uptake by the kidneys. It has been reported that the particle size usually capable of accumulating in the kidneys is roughly between 10 and 75 nm, since particles smaller than 10 nm are quickly clearable, and those greater than 75 nm may barely cross the renal filtration barriers to enter the kidney [24]. Therefore, the micellar size values obtained for TPGS (~14 nm) and PF127 (~40 nm) fit the size range expected for the renal accumulation of these nanocarriers. Regarding TPGS micelles, our results support previous studies that have reported no marked changes in Dh with increases in temperature of up to 40 °C [25]. On the contrary, the micellization of PF127 was affected by changes in temperature. In agreement with other authors, our findings suggest that the increase in temperature causes a decrease in micellar sizes due to the exclusion of the solvent from the micellar core; this results from the enhancement in core hydrophobicity, which further results in more compact micelles [26]. Additionally, as we observed in both cases (TPGS and PF127 micelles), the drug encapsulation usually increases the Dh of polymeric micelles [26,27,28]. TPGS micelles showed a single population size with a narrow distribution that was more homogeneous compared to PF127. The latter showed two different size populations (one of around 40 nm and another below 10 nm), although with a predominance of the larger population. These observations were also reported by other authors [29,30]. The small populations observed that above 1000 nm for both TPGS and PF127 did not correspond to micelles but impurities of the micellar dispersion. The DLS analysis showed that the addition of AEA in both types of micelles did not produce significant changes in the micellar size distribution.

In addition, ZP values indicated that the surface charge, for both the loaded and empty micelles, was approximately neutral, which is usual for nanostructures covered by PEG shells [31]. This is of particular importance since pegylation is another factor that directly influences kidney localization of the nanostructures. In this regard, pegylated nanostructures are considerably retained by kidneys in comparison with the non-pegylated ones [32]. Therefore, the suitable size and pegylation of the polymeric nanomicelles developed here would have allowed for AEA vectorization through localization methods simpler than active targeting, such as passive targeting [5]. This aspect constitutes an outstanding achievement of the present study.

Of particular interest, DLS also showed that, due to its poor water solubility, AEA forms aggregates in an aqueous solution that hinder the pharmacokinetics of this endocannabinoid [33], which does not happen when this active substance is encapsulated in the TPGS or PF127 micelles. Further, the encapsulation of AEA in polymeric nanotransporters confers protection against the destabilizing agents of AEA, such as hydrolysis, enzymes, and temperature, among others [34]. Aside from this, an essential advantage offered by the nanoformulation developed in the present study compared to another previously performed by our group is that, due to its improved characteristics, it can be administered intravenously. Thus, the micellar nanostructure accompanies the drug during its distribution, which does not happen when nanoformulation is administered intraperitoneally. In this latter case, the drug must be released from the polymer matrix that contains it to access each organ or tissue. In this way, it is possible to achieve a longer circulation time and prolonged drug action without jeopardizing its stability [9].

Unlike previous studies that reported TPGS micelles as efficient drug carriers across the blood–brain barrier, our study showed an almost null entry of these micelles into the brain [35]. This result would be especially relevant in our case, where we use AEA as an active ingredient, which can produce specific adverse effects at the level of the CNS, in some circumstances [36]. According to results previously reported by Tesan et al., the accumulation of TPGS micelles was mainly carried out in the liver and kidneys [37,38]. However, unlike one of these studies, where a significant uptake by the lungs was observed, our study showed a low accumulation in these organs, which was replaced by a considerable uptake by the spleen, although it was much smaller than for the liver and kidneys [38]. The biodistribution of PF127 micelles has not been evaluated as such in previous studies, since it has only been determined as mixed micelles prepared in combination with other polymers or PF127 micelles functionalized with some substance for active targeting [39,40]. In this sense, we could see that the pattern of their biological distribution was quite similar to that of TPGS micelles. However, the accumulation in the three main collection organs (liver, kidneys, and spleen) was always a little lower for the micelles of PF127.

It has been shown that several cannabinoids, including AEA, are capable of provoking a triphasic response in the cardiovascular parameters of anesthetized and conscious SHR rats, although with some differences between both groups primarily related to phases II and III [10,41]. In the present study, we only worked with anesthetized rats. Particularly for this group, it has been described that phase I, characterized by abrupt and short bradycardia and hypotension, would be mediated by vanilloid TRPV1 receptors located in sensory vagal nerves along the cardiovascular system. Phase II is associated with a brief pressor response and would be related to tachycardia and an increase in renal and mesenteric blood flow. Finally, phase III, characterized by hypotension, for which the duration is usually 2 to 10 min, would respond to the activation of peripheral CB1 receptors [10,41]. It is noteworthy that our present results with non-encapsulated AEA are following these previous findings. However, of particular interest, this triphasic response has been observed neither with nanoencapsulated AEA (AEA/TPGS and AEA/PF127 micelles) nor with free AEA 24 h after its dispersion in an aqueous solution. On the one hand, this would suggest a hydrolytic degradation of AEA over 24 h is responsible for the disappearance of these effects. On the other hand, this also indicates that TPGS and PF127 micelles would allow a slow and controlled AEA release enough to achieve hypotensive effects while preventing the occurrence of this adverse peripheral response that would be detrimental to cardiovascular homeostasis.

Moreover, it has been reported that AEA—a derivative of arachidonic acid—is highly unstable in the presence of destabilizing environmental factors, such as temperature and oxygen, so this endocannabinoid can easily suffer oxidation and degradation at temperatures above −20 °C [11,42]. Likewise, it is well known that one of the leading causes of chemical instability of active ingredients is hydrolysis [43], so it could be suggested that in our tests, the non-encapsulated AEA also underwent hydrolytic degradation as a result of remaining for 24 h in an aqueous medium. In this sense, the degradation of free AEA in aqueous dispersion could be observed with the naked eye since the freshly prepared dispersion showed whitish turbidity related to the low aqueous solubility of AEA, which ceased to be observed after 24 h, where the aqueous solution was utterly transparent. In the case of the micellar suspension, the AEA aggregate showed no appearance of turbidity, indicating that the endocannabinoid was being encapsulated by the micelles, an aspect that was corroborated by DLS. This fact allowed us to verify that, as another author suggested; encapsulating unstable active ingredients (even under normal environmental conditions) in polymeric nanostructures may enable us to preserve their stability and therapeutic action [40].

To reinforce our present results related to the biological activity of nanoformulated AEA at the renal level, we can find complete agreement with previous in vitro studies carried out both in primary culture of the thick ascending limb of the loop of Henle from Wistar rats, as well as on human proximal tubule (HK2) cell line. More specifically, AEA could exert a blocking and/or reducing activity effect of apical (Na^+^/H^+^ exchanger and Na^+^/K^+^/2Cl^−^ cotransporter) and basolateral (Na^+^/K^+^ ATPase pump) transporters responsible for the reabsorption of sodium at different levels of the nephron [11,44]. Of special interest, the results of the present study show a remarkable difference between AEA encapsulated and non-encapsulated in polymeric matrix related to its ability to increase FES% in SHR animals. This phenomenon could occur because the AEA release from nanomicelles would depend on its diffusion through these polymeric nanostructures and/or micelle disassembly [45]. Therefore, it would be carried out in a gradual and controlled manner compared to the treatment with AEA in a free form [11]. This would support the idea that AEA/TPGS and AEA/PF127 are superior as natriuretic/diuretic systems compared to non-encapsulated AEA since the occurrence of gradual modifications in biological parameters, such as FES% are physiologically “friendlier” than their abrupt changes.

In this context, our study demonstrates, unprecedently, diuretic and natriuretic effects of nanoencapsulated AEA in animals and the use of this endocannabinoid as a new therapeutic alternative for the treatment of hypertension; formulated in such a way that it can be administered in vivo without all disadvantages that limit its use in free form. Further, as mentioned earlier, we observed no hemodynamic, diuretic, or natriuretic effects in WKY rats treated with AEA nanoformulated or not. This finding also matches with other studies that have revealed an increased expression in CB1 receptors at the cardiovascular, and probably, at the renal level in SHR rats compared to their normotensive controls [46]. For this reason, we could explain why AEA did not induce hypotension, diuresis, or natriuresis in normotensive animals. In this regard, more studies should be done to corroborate the differential CB1 expression in the animals used here and to ensure the absence of other factors that could affect the observed behavior.

## 5. Conclusions

Replacing polymeric nanoparticles by polymeric nanomicelles, we significantly reduced the size of AEA-loaded nanostructures to achieve their effective passive accumulation into the kidneys using an administration route transferable to humans. Further, we obtained a nanoformulation with improved steric stability (pegylated nanostructures) utilizing a considerably simpler method of preparation than electrospraying, previously used in the synthesis of other nanoformulations developed by our group. Unpublished, we demonstrated the diuretic, natriuretic and hypotensive effects of nanoformulated AEA in vivo and highlighted the possibility of using AEA as a new therapeutic agent in hypertension treatment, improving its physicochemical, pharmacokinetic and pharmacodynamic limitations. Finally, we determined that both TPGS and PF127 micelles have very similar features and behaviors; however, at our discretion, we consider that TPGS micelles would be a better option in hypertension treatment than PF127 micelles due to their ability to achieve a higher renal accumulation and causing a more effective reduction in blood pressure values.

## Figures and Tables

**Figure 1 pharmaceutics-15-00176-f001:**
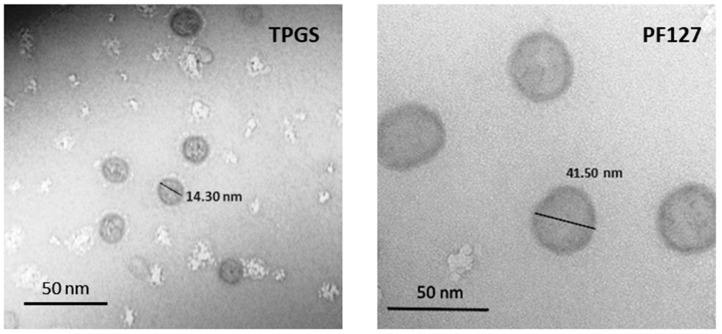
Morphometric characterization of TPGS and PF127 empty micelles: TEM images of TPGS and PF127 micelles showed spherical nanostructures of around 14 and 40 nm, respectively.

**Figure 2 pharmaceutics-15-00176-f002:**
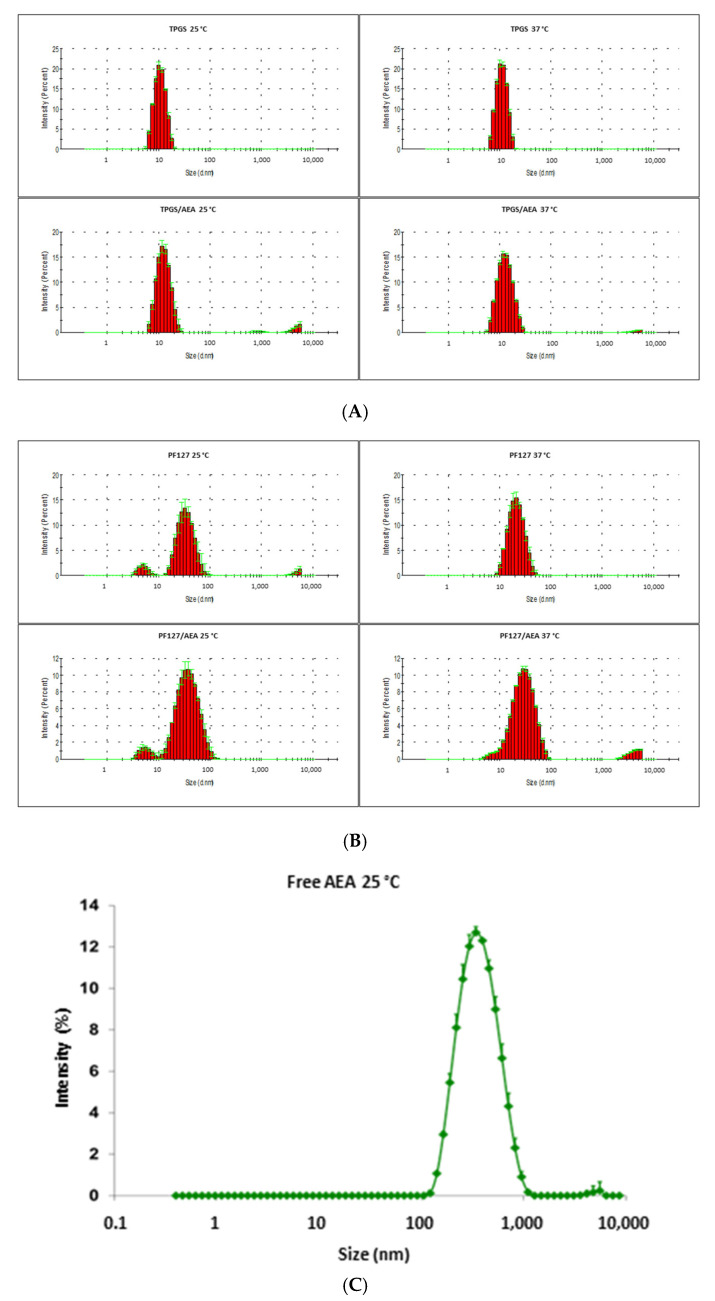
(**A**) Particle size distribution: DLS graphs of TPGS nanomicelles with and without AEA at 25 °C and 37 °C. (**B**) Particle size distribution: DLS graphs of PF127 nanomicelles with and without AEA at 25 °C and 37 °C. (**C**) DLS graphs of free AEA freshly dispersed in aqueous solution at 25 °C.

**Figure 3 pharmaceutics-15-00176-f003:**
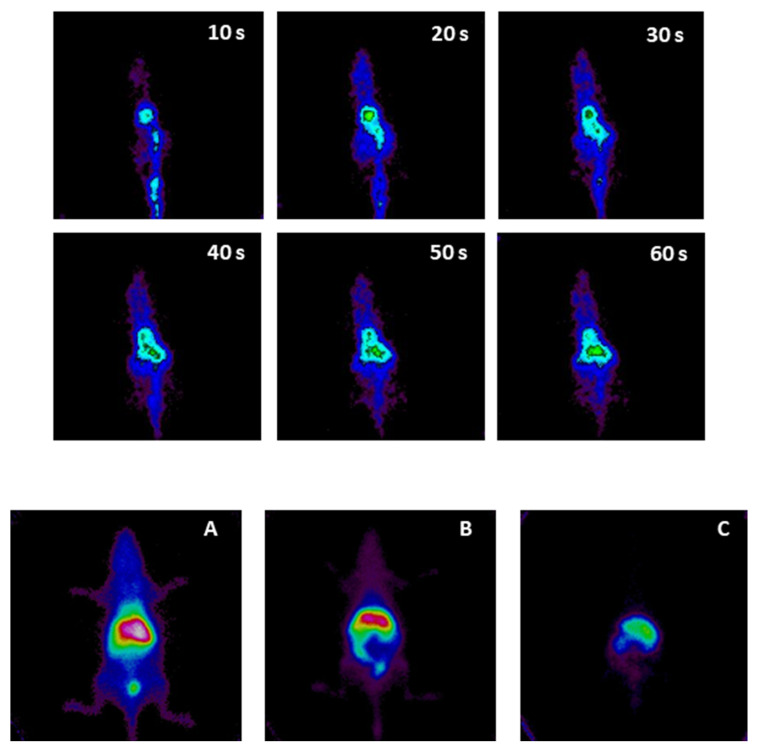
Biodistribution of TPGS micelles during the first minute after intravenous injection, as well as at 30 min (**A**), 3 h (**B**), and 24 h (**C**).

**Figure 4 pharmaceutics-15-00176-f004:**
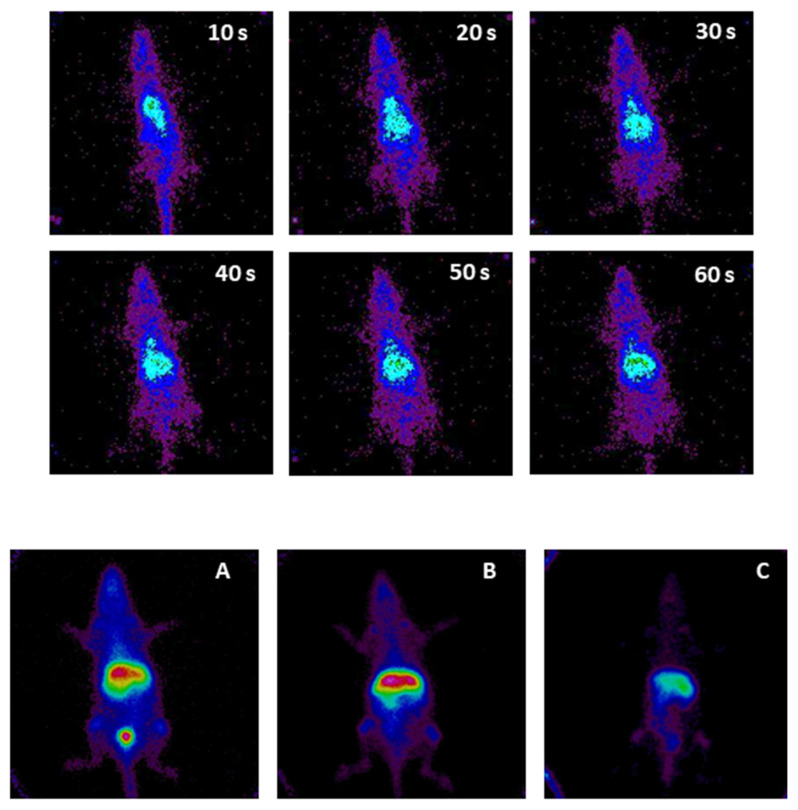
Radiolabeling and gamma camera: biodistribution of PF127 micelles during the first minute after intravenous injection, as well as at 30 min (**A**), 3 h (**B**), and 24 h (**C**).

**Figure 5 pharmaceutics-15-00176-f005:**
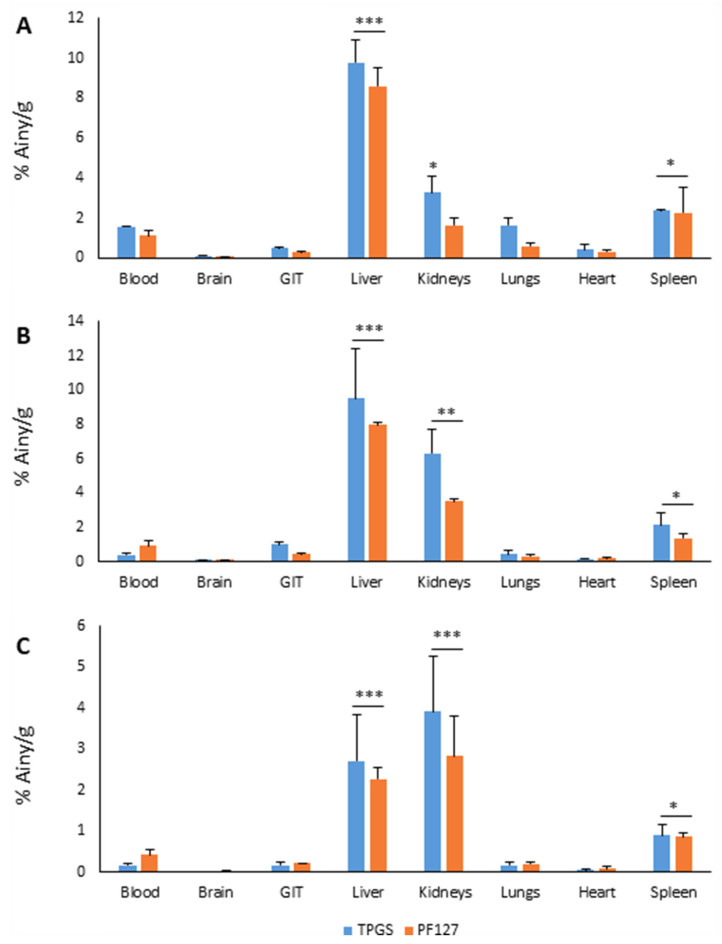
Biodistribution of TPGS (blue) and PF127 (orange) micelles in different organs at 30 min (**A**), 3 h (**B**), and 24 h (**C**) after their intravenous injection. GIT: gastrointestinal tissue. n = 3. * *p* < 0.05; ** *p* < 0.01; *** *p* < 0.001, all versus blood.

**Figure 6 pharmaceutics-15-00176-f006:**
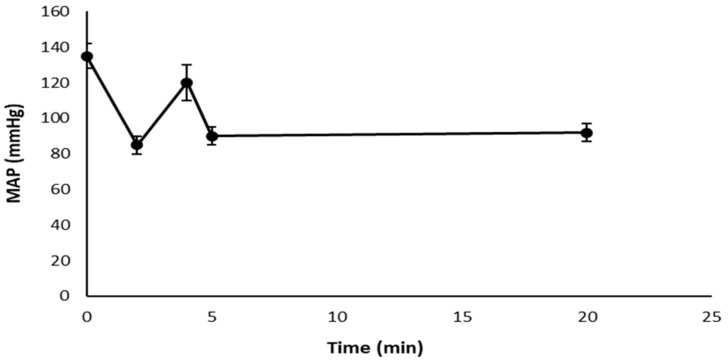
Triphasic response in SHR treated with free AEA freshly dispersed in an aqueous solution. MAP (mmHg) vs. time (min) plot.

**Figure 7 pharmaceutics-15-00176-f007:**
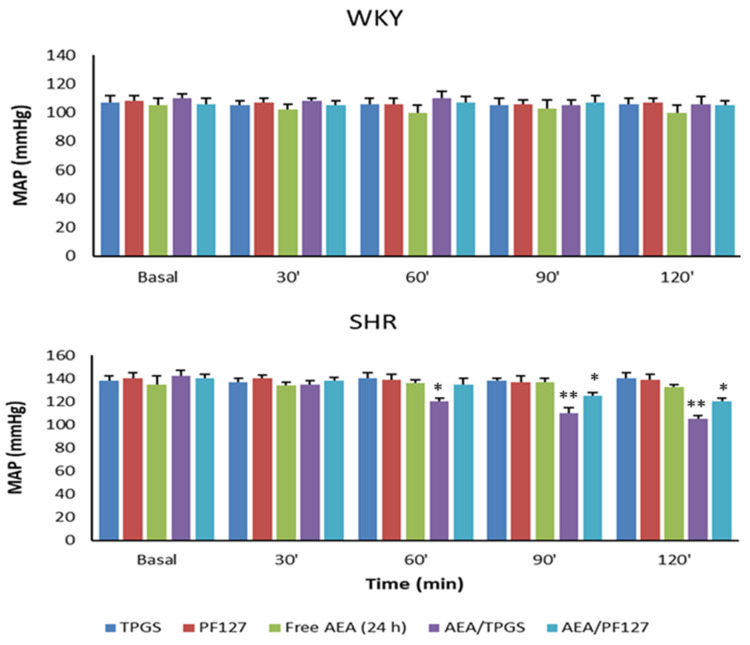
Mean arterial pressure, measured in WKY and SHR rats at basal conditions (control), as well as at 30, 60, 90, and 120 min after injection with different treatments: TPGS empty micelles (TPGS), PF127 empty micelles (PF127), AEA non-encapsulated 24 h after its dispersion in an aqueous solution (free AEA, 24 h), AEA/TPGS micelles (AEA/TPGS), AEA/PF127 micelles (AEA/PF127). n = 3. * *p* < 0.05; ** *p* < 0.01, all versus basal.

**Figure 8 pharmaceutics-15-00176-f008:**
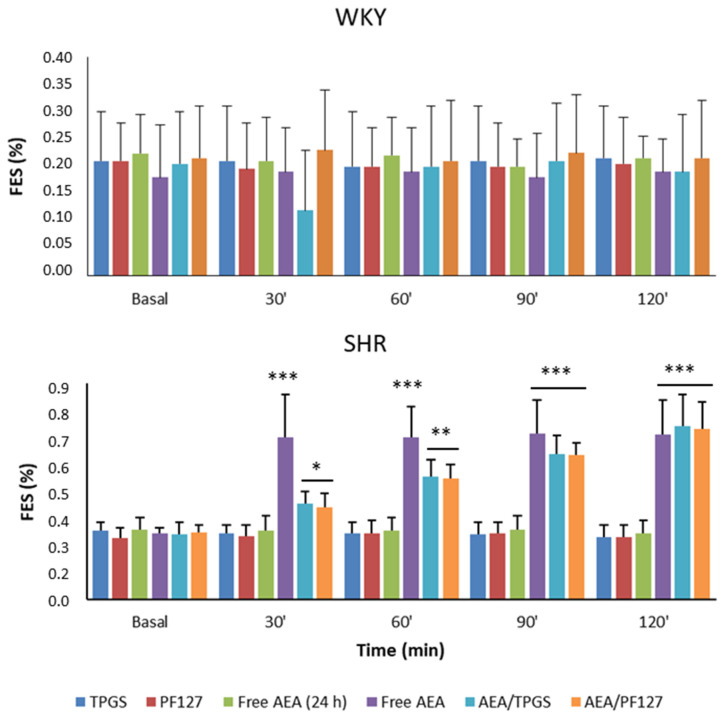
Percentage of the fractional excretion of sodium, measured in WKY and SHR rats, at basal conditions (control) as well as at 30, 60, 90, and 120 min after the injection of different treatments: TPGS empty micelles (TPGS), PF127 empty micelles (PF127), AEA non-encapsulated 24 h after its dispersion in an aqueous solution (free AEA, 24 h), AEA non-encapsulated freshly dispersed in aqueous solution (free AEA), AEA/TPGS micelles (AEA/TPGS), AEA/PF127 micelles (AEA/PF127). n = 3. * *p* < 0.05; ** *p* < 0.01; *** *p* < 0.001, all versus basal.

**Table 1 pharmaceutics-15-00176-t001:** DLS analysis of TPGS and PF127 micelles: The measurements were performed on both empty and AEA-loaded nanomicelles, at 25 °C and 37 °C. n = 3. Dh: average hydrodynamic diameter, ZP: zeta potential, PDI: polydispersity index.

Sample	AEA Presence	Temperature (°C)	Size							ZP (mV)
			Peak 1 (nm)		Peak 2 (nm)		Peak 3 (nm)		PDI (±D.E.)	
			Dh (±D.E.)	%	Dh (±D.E.)	%	Dh (±D.E.)	%		
TPGS 3%		25	10.9 (0.1)	100	-	-	-	-	0.032 (0.013)	−1.455 (0.721)
	√		13.0 (0.4)	100	-	-	-	-	0.234 (0.038)	−2.76 (1.27)
PF127 3%			5.1 (0.5)	8.7	35.4 (3.1)	91.3	-	-	0.391 (0.041)	−1.87 (0.55)
	√		5.8 (0.7)	7.2	40.6 (1.5)	92.8	-	-	0.299 (0.039)	−0.879 (0.414)
TPGS 3%		37	11.2 (0.1)	100	-	-	-	-	0.020 (0.008)	-
	√		13.4 (0.1)	100	-	-	-	-	0.157 (0.008)	-
PF127 3%			22.0 (1.2)	100	-	-	-	-	0.154 (0.039)	-
	√		31.2 (0.3)	100	-	-	-	-	0.287 (0.007)	-

## Data Availability

Not applicable.

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
