# Peer review of "Polymeric Nanomicelles Loaded with Anandamide and Their Renal Effects as a Therapeutic Alternative for Hypertension Treatment by Passive Targeting"

_pharmaceutics, 2023, doi:10.3390/pharmaceutics15010176_

Round 1

Reviewer 1 Report

This manuscript is interesting and novel. However, there is some suggestions:

1. The drug releasing mechanism should be discussed in the manuscript.

2. Please explain the renal accumulation of the PF127. The relative reference should be cited and discussed. 

3. How to prove the complete micelles transportation into kidney, instead of the fragments or polymers of the micelles.

Reviewer 2 Report

In this manuscript, Gimenez and co-workers, described the use of D-α-Tocopheryl polyethylene glycol 1000 succinate (TPGS) micelles and Pluronic® F127 (PF127) polymeric micelles loaded with Anandamide (AEA) as local natriuretic therapeutic delivered to the kidney in order to modulate hypertension in the SHR rat model. 

First they generated TPGS or PF127 micelles and characterized them morphologically by TEM. Then they performed dynamic light scatter measurements to determine the size of the micelles. Subsequently, they determined the biodistribution of the micelles by injecting radiolabeled micelles into rats and measuring solid organds and blood in a scintcounter and found the strongest accumulation in the liver and the kidneys.

When they injected SHR and normotensive WKY rats with TPGS+AEA or PF127+AEA they measured a significant drop of MAP after 60 minutes with TPGS+AEA and 90minutes with PF127+AEA. Measuring FES they found an increase in sodium excretion in rats injected with TPGS+AEA or PF127+AEA, but a much stronger effect using unbound AEA. 

The study is of interest, and the experiments have been carried out well. I don’t see a technical problem with this study. However, the fact that unbound AEA seems to exert a stronger effect natriuretic effect sooner then TPGS+AEA or PF127+AEA needs an explanation before this study can be published.

Major:

- The authors completely missed to explain the findings of Figure 8 in the manuscript. I can only assume this is because it didn’t support the claim that the use of TPGS+AEA or PF127+AEA is superior then non-targeted delivery.

The authors should describe their findings in the text, and add a discussion about the unclear finding. 

Minor:

- please add the written-out form for SHR, in the abstract

- please state that animal procedures have been carried out according to nationally established standards, and that animal experiments have been reviewed by ethic committee, also add protocol number of the approved animal protocol

- please describe the method of euthanasia

Reviewer 3 Report

The authors described the effect of AEA-loaded nanomicelles as a therapeutic alternative for hypertension treatment by passive targeting. This manuscript can be considered for publication once the authors address the following queries:

In the preparation of AEA-loaded micelles, you better determine the drug-loading content and encapsulation efficiency.

I was wondering how do you determine the concentration of AEA in AEA/PF127 and AEA/TPGS for application purpose.

The size will affect the biodistribution and bioavailability of AEA-loaded nanomicelles. What is the ideal size of AEA/PF127 and AEA/TPGS for this application?

To elicit the intended activity, AEA should be released from the micelle's core once it reaches its target. So, what is the releasing stimuli for AEA? What are the concentrations of the stimuli?

I believe in vitro AEA releasing study should be conducted by employing the releasing stimuli (for both AEA/PF127 and AEA/TPGS).  

The stability of AEA/PF127 and AEA/TPGS is essential to curb the premature release of AEA in circulation. Thus, in vitro stability studies against dilution and serum proteins should be conducted.

What was the concentration of AEA used in experiments which generated the data included in Figure 5-8. How do you calculate the concentration of AEA in these experiments?

A biocompatibility study should be conducted for AEA/PF127 and AEA/TPGS using normal cell lines   

Round 2

Reviewer 2 Report

The authors addressed the critical points and improved the manuscript.  I recommend publication.

However, the authors should removed the word "slaughtered" in line 211 and replace it with "sacrificed".

Reviewer 3 Report

The authors responses about my concerns are not satisfactory.